# An Accurate Multiple Sclerosis Detection Model Based on Exemplar Multiple Parameters Local Phase Quantization: ExMPLPQ

**Gulay Macin** [1], **Burak Tasci** [2], **Irem Tasci** [3], **Oliver Faust** [4,*], **Prabal Datta Barua** [5,6], **Sengul Dogan** [7], **Turker Tuncer** [7], **Ru-San Tan** [8,9] and **U. Rajendra Acharya** [10,11,12]

1 Department of Radiology, Beyhekim Training and Research Hospital, Konya 42001, Turkey; gulaymacin@gmail.com
2 Vocational School of Technical Sciences, Firat University, Elazig 23119, Turkey; btasci@firat.edu.tr
3 Department of Neurology, School of Medicine, Malatya Turgut Ozal University, Malatya 44100, Turkey; irem.tasci@ozal.edu.tr
4 Department of Engineering and Mathematics, Sheffield Hallam University, Sheffield S1 1WB, UK
5 School of Business (Information System), University of Southern Queensland, Toowoomba, QLD 4350, Australia; prabal.barua@usq.edu.au
6 Faculty of Engineering and Information Technology, University of Technology Sydney, Sydney, NSW 2007, Australia
7 Department of Digital Forensics Engineering, Technology Faculty, Firat University, Elazig 23119, Turkey; sdogan@firat.edu.tr (S.D.); turkertuncer@firat.edu.tr (T.T.)
8 Department of Cardiology, National Heart Centre Singapore, Singapore 599489, Singapore; tanrsnhc@gmail.com
9 Duke-NUS Medical School, Singapore 169857, Singapore
10 Ngee Ann Polytechnic, Department of Electronics and Computer Engineering, Singapore 599489, Singapore; aru@np.edu.sg
11 Department of Biomedical Engineering, School of Science and Technology, SUSS University, Singapore 599491, Singapore
12 Department of Biomedical Informatics and Medical Engineering, Asia University, Taichung 413, Taiwan
* Correspondence: oliver.faust@gmail.com

**Abstract:** Multiple sclerosis (MS) is a chronic demyelinating condition characterized by plaques in the white matter of the central nervous system that can be detected using magnetic resonance imaging (MRI). Many deep learning models for automated MS detection based on MRI have been presented in the literature. We developed a computationally lightweight machine learning model for MS diagnosis using a novel handcrafted feature engineering approach. The study dataset comprised axial and sagittal brain MRI images that were prospectively acquired from 72 MS and 59 healthy subjects who attended the Ozal University Medical Faculty in 2021. The dataset was divided into three study subsets: axial images only ($n$ = 1652), sagittal images only ($n$ = 1775), and combined axial and sagittal images ($n$ = 3427) of both MS and healthy classes. All images were resized to 224 × 224. Subsequently, the features were generated with a fixed-size patch-based (exemplar) feature extraction model based on local phase quantization (LPQ) with three-parameter settings. The resulting exemplar multiple parameters LPQ (ExMPLPQ) features were concatenated to form a large final feature vector. The top discriminative features were selected using iterative neighborhood component analysis (INCA). Finally, a k-nearest neighbor (kNN) algorithm, Fine kNN, was deployed to perform binary classification of the brain images into MS vs. healthy classes. The ExMPLPQ-based model attained 98.37%, 97.75%, and 98.22% binary classification accuracy rates for axial, sagittal, and hybrid datasets, respectively, using Fine kNN with 10-fold cross-validation. Furthermore, our model outperformed 19 established pre-trained deep learning models that were trained and tested with the same data. Unlike deep models, the ExMPLPQ-based model is computationally lightweight yet highly accurate. It has the potential to be implemented as an automated diagnostic tool to screen brain MRIs for white matter lesions in suspected MS patients.

**Keywords:** multiple sclerosis; magnetic resonance imaging; feature engineering; local phase quantization

## 1. Introduction

Multiple sclerosis (MS) is a chronic autoimmune inflammatory disease characterized by demyelination and axonal loss in central nervous system neurons of the brain, spine, and optic nerves [1] that affects approximately 2.8 million people worldwide [2]. There are secular trends of increasing disease incidence and prevalence that are non-homogeneously distributed across populations due to complex gene-gene and gene-environmental interactions and increasing incidence among females. Uncertainties regarding the ascertainment of MS diagnosis and censoring of survival data after diagnosis may confound accurate estimations of incidence and prevalence rates, respectively [3]. Notwithstanding this, it remains indisputable that MS has exacted high economic costs on the healthcare systems of both developed [4] and low- to middle-income countries [5]. The main drivers of healthcare resource utilization are the costs of disease-modifying therapies (DMTs) and the non-medical costs associated with the management of chronic disability in the early and advanced stages of the disease, respectively [4,5]. While patients and their families frequently bear the costs of non-medical interventions, these interventions are nevertheless associated with increased long-term medical and total societal costs [4]. This underscores the need for early diagnosis and intervention with DMTs, which can potentially control disease progression [1,6,7] and have been shown to be cost-effective in improving patients' quality of life [8]. This provides ample justification and motivation for the ongoing quest for more sensitive and accurate methods of MS diagnosis.

There is no single pathognomonic clinical or laboratory finding that can secure a definitive diagnosis of MS. Rather, the diagnosis is made based on consensus clinical, imaging, and laboratory criteria [1,6]. The 2017 McDonald criteria [9] define typical clinical signs and symptoms as well as lesions on magnetic resonance imaging (MRI) [10] that manifest in time and space, which can be combined with auxiliary examination findings (cerebrospinal fluid examination, visual and somatosensory evoked potentials) to establish the diagnosis of MS. Of note, MRI plays an instrumental role in the identification and localization of characteristic demyelinating plaques in the white matter of the brain and spine that constitute the pathological basis of MS and underpin its neurological presentations [10–12]. In particular, T2-weighted fluid-attenuated inversion recovery (FLAIR) MRI [13] offers the optimal contrast-to-noise image signal properties for sensitive detection of plaque lesions and is routinely performed for anatomical MRI screening of the central nervous system in suspected cases of MS [12]. Interpretation of MRI images requires experts to manually scrutinize multiple contiguous image sections for the presence of white matter lesions, with care being taken to distinguish MS plaques from lesions associated with diseases that present with similar symptoms, e.g., ischemic gliosis and central nervous system vasculitis [14–16]. MS plaques are hyperintense in T2 sequence, oval-round shaped, at least 3 mm in size, with asymmetric distribution. The typical location of MS plaques is as follows: (1) periventricular: adjacent to the lateral ventricles; (2) juxtracortical and cortical: localized to U fibers; (3) infratentorial: located unilaterally or bilaterally paramedian adjacent to the brain stem, cerebellum, and 4th ventricle; (4) spinal cord: cervical and thoracic localized, shorter than two vertebral segments, axially wedge-shaped, sagittal cigarette shaped, localized in peripherally located posterior and lateral columns [12]. The process of clinical scan reading is time-intensive, fatiguing, and susceptible to intra- and inter-observer variability. These limitations provide an opportunity for harnessing the power of artificial intelligence (AI) for screening large numbers of MRI images to detect MS, which can be posed as a problem of classifying images with and without white matter lesions [17]. AI mimics human intelligence to perform tasks and can progressively attain higher accuracy by collecting more information [18]. These desirable traits have spurred the adoption of AI methods in many healthcare applications, which can potentially ease the workload of



medical and paramedical personnel. To this end, AI has been used for computer-aided MS diagnosis [19,20] and prognostication of disease progression [19,20]. Accurate AI-enabled MS detection promises earlier diagnosis and treatment initiation with DMTs, better disease surveillance, and more efficient utilization of healthcare resources. However, questions remain about the reliability and practicality of AI-enabled MS detection.

Some MS detection studies presented in the literature are given below.

Storelli et al. [21] analyzed the MRIs of 373 MS patients using a CNN model and attained accuracy rates of 83.3%, 67.7%, and 85.7% for clinical, cognitive, and combined clinical plus cognitive diagnoses, respectively. The parameter values and optimization methods used in the CNN architecture negatively affected the classification results. Alijamaat et al. [22] proposed a method that incorporated a two-dimensional discrete Haar wavelet transform and CNN to study the MRIs of 38 patients and 20 healthy individuals and attained sensitivity, specificity, precision, and accuracy of 99.14%, 98.89%, 99.43%, and 99.05%, respectively, in their experiments. Oliveira et al. [23] proposed a method for measuring plaque volume using MRIs from four different datasets. Their proposed method achieved 99.69%, 98.51%, 98.51%, and 99.85% accuracy, precision, sensitivity, and specificity. Narayana et al. [24] studied T1, T2, and FLAIR MRIs of 489 healthy and 519 MS patients. Using the Vgg16+FCN network structure, they attained 72%, 70%, and 70% sensitivity, specificity, and accuracy. Barquero et al. [25] studied MRIs of 124 MS patients. Using a Rim-Net CNN architecture, they attained 62.3%, 75.8%, 95.1%, and 93.8% F1-score, sensitivity, specificity, and accuracy. Ye et al. [26] used diffusion-based spectrum imaging techniques to study the MRIs of 38 MS patients. Using a deep neural network, they attained 97.3%, 99.1%, 97.3%, and 93.4% F1-score, sensitivity, specificity, and accuracy. Vogelsanger and Federau [27] studied a large dataset of 1855 healthy MRIs, 2910 MRIs from 616 MS patients, and 639 MRIs from 625 leukoencephalopathy patients attained precision and recall rates of 92% and 89%, respectively. Shrwan and Gupta [28] studied the MRIs of 38 MS patients using a 2D-CNN network and attained 99.55%, 99.15%, and 99.15% accuracy, precision, and recall, respectively. Afzal et al. [29] used 127 scans from the Medical Image Computing and Computer-Assisted Intervention 2016 and International Symposium on Biomedical Imaging 2015 datasets in their study. In their segmentation study, they attained 67%, 48%, and 90% dice similarity coefficient, sensitivity, and precision. Afzal et al. [30] conducted two experiments using MRIs of 21 MS patients using a CNN network model and attained 83.3% and 100% accuracy rates in the first and second experiments, respectively.

The datasets in the studies presented above [22,28–30] are small. In Storelli et al. [21] and Narayana et al. [24], the dataset is large, but has a low accuracy rate. In [21–28,30], computational complexity is high. With our work on an accurate MS detection model, we attempt to address some of the issues and problems raised above. Our solution took the form of a reliable machine learning model for classifying FLAIR images of the brain into MS and non-MS classes accurately. To this end, we created a novel exemplar feature engineering algorithm based on local phase quantization (LPQ), which we named exemplar multiple parameters LPQ (ExMPLPQ). The model was created by fusing ExMPLPQ with a machine learning algorithm and training and testing it on a prospectively acquired brain MRI data set. The model comprised four phases: (1) brain MRI image segmentation; (2) exemplar feature extraction using LPQ; (3) feature selection using iterative neighborhood component analysis (INCA); and (4) classification using shallow k-nearest neighbor (kNN) classifier. The contributions of our work are as follows:

- A prospective brain MRI dataset was collected to train and test the proposed ExMPLPQ model. The dataset has been made publicly available.
- The handcrafted ExMPLPQ model attained over 97% classification accuracy on the study dataset. In addition, our results for MS detection were demonstrably superior to 19 state-of-the-art pretrained methods, which included transfer learning and deep learning models.

## 2. Materials and Methods

This section describes the study dataset, proposed method, including feature extraction and classification process.

### 2.1. Materials

The study dataset comprised axial and sagittal FLAIR MRI images of the brain that were prospectively acquired from 72 MS and 59 non-diseased "healthy" male and female patients who attended the Ozal University Medical Faculty in 2021. The institutional ethics committee had approved the study. Medical experts read the FLAIR image sections. From the 72 MS patients, 1441 axial and sagittal brain image sections containing identifiable MS lesions were assigned to the MS class; and from the 59 non-diseased patients, 2016 axial and sagittal images sections with normal appearance, i.e., without white matter lesions, were assigned to the healthy class (Table 1). For binary classification into MS vs. healthy, three study data subsets comprising axial only (*n* = 1652) and sagittal only (*n* = 1775) (Figure 1), and combined axial and sagittal images were created (*n* = 3427) (Table 1). The dataset can be downloaded at: https://www.kaggle.com/datasets/buraktaci/multiple-sclerosis (accessed on 7 April 2022).

**Table 1.** The attributes of the MRI dataset used.

| | Male, *n* | Female, *n* | Total, *n* | Age, Years | Number of MRI Images, *n* |
|---|---|---|---|---|---|
| MS-Axial | 21 | 51 | 72 | 28.4 ± 5.66 | 650 |
| MS-Sagittal | 21 | 51 | 72 | 28.4 ± 5.66 | 761 |
| Healthy-Axial | 27 * | 30 * | 57 * | 29.5 ± 8.32 | 1002 |
| Healthy-Sagittal | 29 * | 20 * | 49 * | 27.4 ± 6.48 | 1014 |

* There is an overlap of subjects in the healthy class, which comprises 29 males and 30 females.

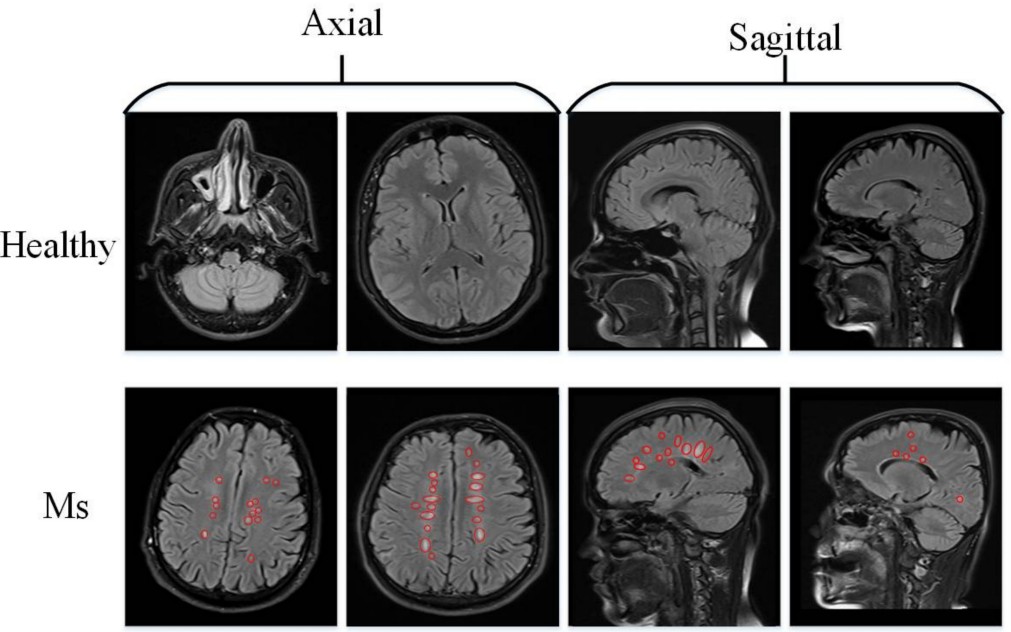

**Figure 1.** Example axial and sagittal FLAIR MRI sections in healthy and multiple sclerosis (MS) classes. Note the presence of hyperintense lesions in the brain's white matter in the latter.

### 2.2. Transfer Learning-Based Feature Engineering Model

The ExMPLPQ model combines desirable properties of feature extraction based on exemplar and multiple parameters. LPQ [31], a popular textural feature extractor, was deployed to generate textural features. These were fed to an INCA selector to select the top

discriminative features. A kNN classifier [32] was employed for MS vs. non-MS classes (Figure 2).

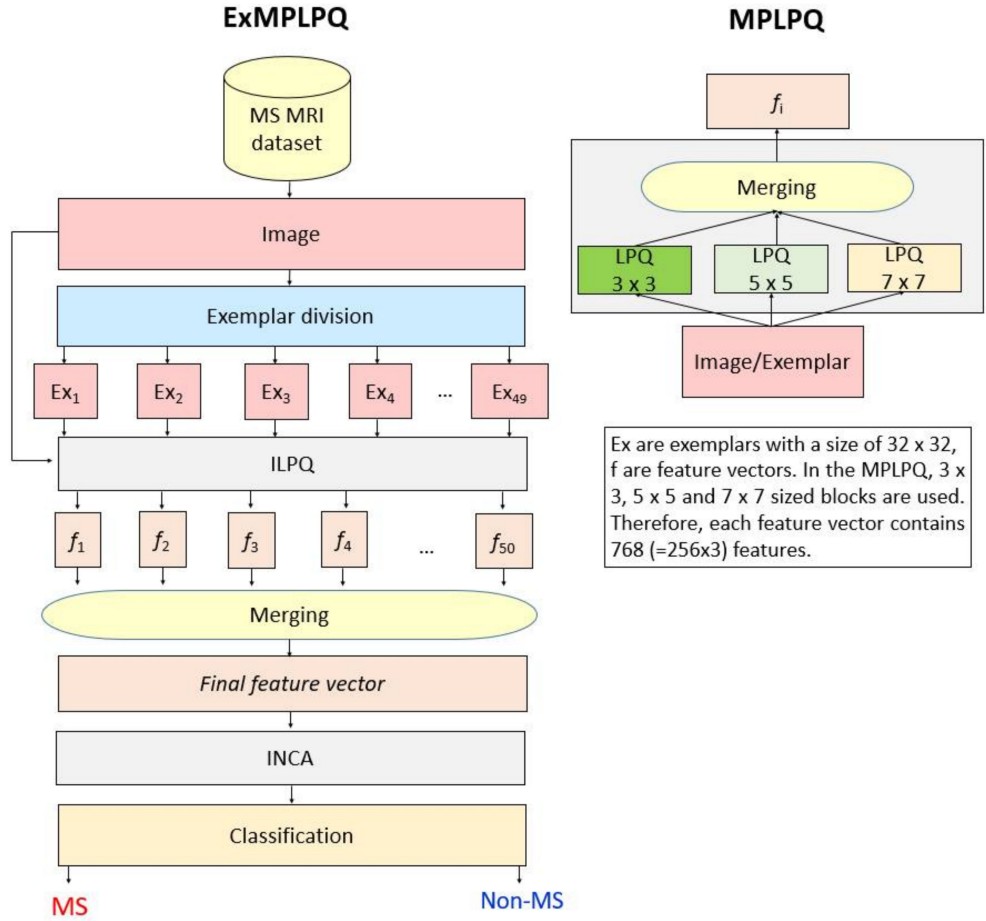

**Figure 2.** ExMPLPQ flow diagram.

The pseudocode of the proposed method was presented in Algorithm 1.

| **Algorithm 1.** Detailed flow of the ExMPLPQ technique |
|---|
| Input: The used MS dataset with 3.427 MRIs. |
| Output: Results. |
| 00: Load MS dataset |
| 01: **for** k = 1 to 3427 **do** |
| 02:         I = MS(k); Image reading from MS dataset |
| 03:         Reshape image into 224 × 224 sized image |
| 04:         X(k,1:768) = [lpq(I,3) lpq(I,5) lpq(I,7)] |
| 05:           **for** i = 1 to 224 step by 32 **do** |
| 06:             **for** j = 1 to 224 step by 32 **do** |
| 07:                 exm = res(i:i+31,j:j+31,1) |
| 08:                 X(k, counter * 768+1:(counter+1) * 768) = [lpq(exm,3) lpq(exm,5) lpq(exm,7)] |
| 09:             **end for j** |
| 10:           **end for i** |
| 11:         Extract the 50th feature vector using the resized images. |
| 12:         Merge the generated feature vectors to create the final feature vector. |
| 13: **end for k** |
| 14: Apply INCA to generated features. |
| 15: Forward the selected features to kNN classifiers. |
| 16: Obtain predict values |

The processing steps of the ExMPLPQ algorithm are described below.

*Step 1:* Read each image from the collected MRI study data subsets.

*Step 2:* Resize each MRI image to 224 pixels × 224 pixels.

*Step 3:* Divide each resized image into 49 (=7 × 7) 32 × 32 sized patches/exemplars.

$$Ex_t = Im(ii + 32 \times (i-1), jj + 32 \times (j-1)), \ i \in \{1, 2, \ldots, 7\}, \ j \in \{1, 2, \ldots, 7\}, \\ t \in \{1, 2, \ldots, 49\}, \ ii \in \{1, 2, \ldots, 32\}, \ jj \in \{1, 2, \ldots, 32\} \tag{1}$$

where $Ex_t$ represents the $t^{\text{th}}$ fixed-size patch with size 32 × 32, and $Im$ is the resized image with size 224 × 224.

*Step 4:* Generate features by applying the LPQ feature extractor function.

$$feat_1 = LPQ(Ex^t, 3 \times 3), \ t \in \{1, 2, \ldots, 49\} \tag{2}$$

$$feat_2 = LPQ(Ex^t, 5 \times 5) \tag{3}$$

$$feat_3 = LPQ(Ex^t, 7 \times 7) \tag{4}$$

$$f_t = conc(feat_1, feat_2, feat_3) \tag{5}$$

where $feat$ represents LPQ features generated by deploying $LPQ(.,.)$ function with blocks of varying sizes; and $conc(.,.,.)$ concatenation function. Each of the three $feat$ has 256 features, which are concatenated to form $f_t$ with a length of 768.

$$feat_1 = LPQ(Im, 3 \times 3), \ t \in \{1, 2, \ldots, 49\} \tag{6}$$

$$feat_2 = LPQ(Im, 5 \times 5) \tag{7}$$

$$feat_3 = LPQ(Im, 7 \times 7) \tag{8}$$

$$f_{50} = conc(feat_1, feat_2, feat_3) \tag{9}$$

*Step 5:* Extract the 50th feature vector using the resized images.

*Step 6:* Merge the generated feature vectors ($f$) to create the final feature vector.

$$ftv = (j + 768 \times (i-1)), j \in \{1, 2, \ldots, 768\}, \ i \in \{1, 2, \ldots, 50\} \tag{10}$$

where $ftv$ represents the final feature vector with a length of 38,400 (=768 × 50) generated from each MRI image.

The next processing steps involve the INCA feature selection function, during which the algorithm selects 403, 716, and 944 features for the axial, sagittal, and hybrid MRI study data subsets, respectively.

*Step 7:* Calculate indexes sorted by applying the neighborhood component analysis (NCA) [33] selector.

$$ind = \psi(ftv, y) \tag{11}$$

where $\psi$ represents the NCA feature selection function; $ind$, indexes qualified according to distinctiveness; and $y$, actual labels.

*Step 8:* Apply iterative feature selection using the calculated indexes ($ind$). Moreover, loss values of each selected feature vector deploying the kNN classifier. In this work, 901 feature vectors (the used iteration range is from 100 to 1000) have been selected, and kNN calculates each feature vector's loss/misclassification rate to choose the best/optimal feature vector. This process is given in below.

$$l^i = \kappa(ftv(:, ind(t)), y, kf), \ t \in \{1, 2, \ldots, i+99\}, \ i \in \{1, 2, \ldots, 901\} \tag{12}$$

where $l^i$ represents the loss value of the selected $i^{\text{th}}$ feature vector; and $\kappa$, the kNN classifier. $\kappa$ incorporates three parameters: feature vector, actual output ($y$), and validation ($kf$). In this work, $kf$ is chosen as 10-fold cross-validation.

*Step 9:* Select the best feature vector using the calculated loss values in Step 8.

$$[mini, idx] = \min(l) \tag{13}$$

$$final = ftv(:, ind(g)), \ g \in \{1, 2, \dots, idx + 99\} \tag{14}$$

where *final* represents the selected best feature vector; *mini* are the minimum loss values; and *idx*, are the indexes of these (minimum) values. In these equations (see Equations (13) and (14)), the index (*idx*) of the feature vector with has a maximum classification ratio. In Equation (13), the index of the best feature vector is calculated, and this feature vector (final) is selected using Equation (14).

In the last step, classification is performed.

*Step 10:* Classify the chosen final feature vector (*final*) by deploying a shallow kNN algorithm, Fine kNN [34], using 10-fold cross-validation. Of note, Fine kNN has been used for loss/misclassification rate calculations during INCA feature selection and binary classification. The hyperparameters were set to k is one; distance function, Spearman; and voting, none.

As can be seen from these 10 steps above, the proposed local phase quantization-based MR image classification is a parametric method. The used parameters in this work are tabulated in Table 2.

**Table 2.** The used parameters.

| Step | Parameter |
|---|---|
| Exemplar division | $32 \times 32$ sized patches have been used. To generate features from local areas. |
| LPQ | This step was used to extract textural features with variable parameters (we have used 3, 5, and 7 parameters). Therefore, this feature extractor is named MPLPQ. The prime purpose of this feature extractor is to use the effectiveness of the LPQ by using variable parameters. |
| Feature concatenation | The proposed model extracts 768 features from each patch and raw image. In our architecture, 50 (=49 + 1) feature vectors have been created, and each feature vector has 768 features. Therefore, the created feature vector has 38,400 (=768 $\times$ 50) features. |
| INCA | Loop range: from 100 to 1000<br>Loss vector: kNN |
| Classification using kNN | k: 1<br>Distance: Spearman<br>Voting: None |

As shown in Table 2, our proposed architecture is mimicked by a deep model, but we have used a handcrafted features-based model. We have selected the size of the patch as $32 \times 32$ to extract features from local areas, and we have used an effective feature extractor. This feature extractor generates textural features from both the space and frequency domain. Moreover, it uses parameters. We have used three parameters to use the effectiveness of them together.

## 3. Performance Analysis

MATLAB 2021b was used to implement the ExMPLPQ model algorithms. The implementation was structured with a set of modular functions (main, pre-processor (exemplar division function), Inception local phase quantization (ILPQ), INCA, and kNN). The model was trained and tested with three study data subsets comprising axial images only, sagittal images only, and combined axial and sagittal images. The model performance was compared against 19 pre-trained models, in which kNN was deployed as a classifier, and

the classification was repeated 100 times for each pre-trained model. The ExMPLPQ algorithm possessed low computational complexity and was executed on a desktop personal computer with Windows 10.1 pro-OS, 11th generation Intel i9 processor, 32 GB memory, and 1.5 TB hard disk without parallel processing or graphics processing. Standard performance metrics—precision, sensitivity, specificity [35,36], F1-score, accuracy, Matthew's correlation coefficient (MCC) [35,36]—were used to evaluate ExMPLPQ as well as the 19 pre-trained models.

## 4. Results

Our model attained excellent binary classification performance with >97% accuracy and >95% performance across all standard evaluation metrics (Table 3), as well as relatively low rates of misclassification in all three study data subsets (Figure 3).

**Table 3.** Calculated performance results (%) per classifier used.

| Data Subset | Accuracy | Sensitivity | Specificity | Precision | F-Score | MCC |
|---|---|---|---|---|---|---|
| Axial | 98.37 | 96.46 | 99.60 | 99.37 | 97.89 | 96.59 |
| Sagittal | 97.75 | 95.01 | 99.80 | 99.72 | 97.31 | 95.46 |
| Hybrid | 98.22 | 96.39 | 99.50 | 99.27 | 97.81 | 96.34 |

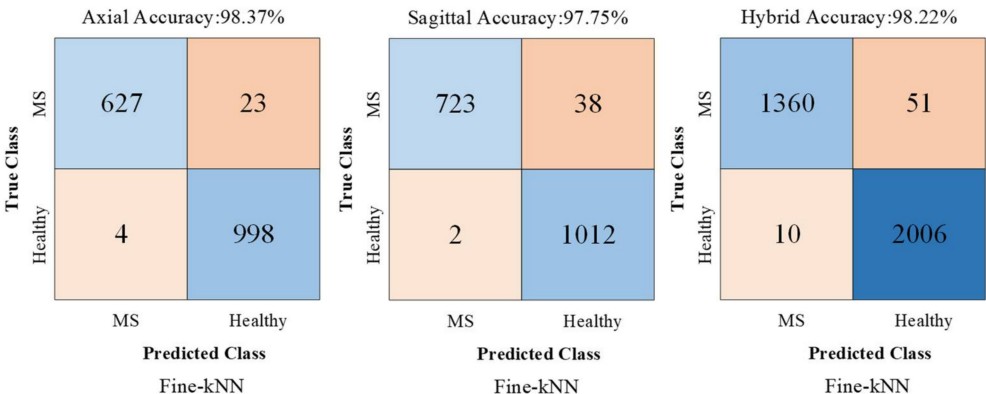

**Figure 3.** Confusion matrices for binary classification using Fine kNN classifier.

The proposed model was run 100 times for axial, sagittal, and hybrid data using a Fine kNN classifier with 10-fold cross-validation. The mean ± standard deviations are tabulated in Table 4.

**Table 4.** The general classification results (%) using a Fine kNN classifier with 10-fold cross-validation ± standard deviations.

| Data Subset | Accuracy | Sensitivity | Specificity | Precision | F-Score | MCC |
|---|---|---|---|---|---|---|
| Axial | 98.35 ± 0.02 | 96.45 ± 0.03 | 99.59 ± 0.01 | 99.36 ± 0.02 | 97.88 ± 0.02 | 96.58 ± 0.04 |
| Sagittal | 97.74 ± 0.019 | 95.00 ± 0.044 | 99.79 ± 0.054 | 99.71 ± 0.075 | 97.30 ± 0.024 | 95.45 ± 0.03 |
| Hybrid | 98.20 ± 0.06 | 96.38 ± 0.06 | 99.49 ± 0.06 | 99.26 ± 0.04 | 97.80 ± 0.05 | 96.32 ± 0.012 |

MCC refers to Matthew's correlation coefficient.

## 5. Discussion

Since the 2000s, deep learning AI techniques have become very popular for diverse applications due to their high performance [37–39]. They have been applied in the biomedical field with notable success [40–43]. Some of these algorithms can veritably be implemented in the clinical environment as medical decision support systems to assist physicians and/or paramedical personnel. Unfortunately, deep learning models are computationally intensive

and demand high time costs for parameter tuning. This study proposed a computationally lightweight machine learning model with handcrafted feature engineering for diagnosing MS on FLAIR brain images. The performance of the proposed model was compared with the results of 19 pre-trained models, including deep learning and transfer learning models, that were also trained and tested on a common prospectively acquired brain FLAIR MRI dataset comprising three subsets of images in different orientations. Fixed-sized patches with a size of $32 \times 32$ have been used to extract deep features using transfer learning. After extracting features from individual images in the study data subsets using the ExMPLPQ algorithm and the 19 pre-trained models, kNN classifier with 10-fold cross-validation was used to perform binary classification of the input images into MS vs. non-MS classes. The proposed ExMPLPQ attained excellent performance with >97% accuracy across all three study data subsets (Table 2). In contrast, the accuracy results after 100 classification runs of the 19 pre-trained models, which incorporated between 1.24 million and 144 million parameters, were all inferior to our model (Table 4). Among the 19 pre-trained models, Efficient b0 had the highest accuracy rates of 93.69%, 90.26%, and 93.22% for the axial, sagittal, and hybrid images, respectively. In contrast, GoogleNet attained the lowest corresponding accuracy rates of 84.49%, 82.02%, and 85.62. Table 5 shows comparative results (classification accuracies) for transfer learning methods.

**Table 5.** Means $\pm$ standard deviations of accuracy results attained by the19 comparator pre-trained models after 100 classification runs using Fine kNN classifier with 10-fold cross-validation.

| Number | Pre-Trained Model | Axial Accuracy | Sagittal Accuracy | Hybrid Accuracy |
|---|---|---|---|---|
| 1 | GoogleNet [44] | 84.49 ± 0.53 | 82.02 ± 0.58 | 85.62 ± 0.35 |
| 2 | DarkNet53 [45] | 87.79 ± 0.47 | 86.49 ± 0.63 | 88.02 ± 0.32 |
| 3 | Inceptionv3 [46] | 89.06 ± 0.48 | 82.47 ± 0.62 | 88.19 ± 0.33 |
| 4 | NasnetLarge [47] | 86.20 ± 0.41 | 81.63 ± 0.52 | 88.22 ± 0.30 |
| 5 | NasnetMobile [47] | 87.48 ± 0.34 | 82.41 ± 0.56 | 88.56 ± 0.30 |
| 6 | VGG19 [48] | 87.80 ± 0.60 | 83.03 ± 0.56 | 88.58 ± 0.37 |
| 7 | VGG16 [48] | 88.54 ± 0.45 | 84.78 ± 0.54 | 88.79 ± 0.31 |
| 8 | Resnet101 [49] | 88.76 ± 0.47 | 85.86 ± 0.51 | 88.90 ± 0.32 |
| 9 | Inceptionresnetv2 [50] | 90.10 ± 0.38 | 84.18 ± 0.55 | 89.42 ± 0.27 |
| 10 | AlexNet [51] | 87.38 ± 0.57 | 84.57 ± 0.51 | 89.77 ± 0.34 |
| 11 | ShuffleNet [52] | 90.25 ± 0.54 | 86.25 ± 0.54 | 90.12 ± 0.35 |
| 12 | Resnet50 [49] | 90.81 ± 0.51 | 88.33 ± 0.46 | 90.15 ± 0.35 |
| 13 | Xception [53] | 91.35 ± 0.44 | 86.15 ± 0.51 | 90.24 ± 0.28 |
| 14 | Resnet18 [49] | 91.50 ± 0.42 | 85.77 ± 0.48 | 90.45 ± 0.32 |
| 15 | Darknet19 [45] | 89.90 ± 0.51 | 85.61 ± 0.54 | 90.57 ± 0.31 |
| 16 | MobileVnet2 [54] | 91.08 ± 0.46 | 85.70 ± 0.50 | 91.15 ± 0.31 |
| 17 | DenseNet201 [55] | 91.88 ± 0.53 | 87.75 ± 0.50 | 91.81 ± 0.30 |
| 18 | SqueezeNet [56] | 90.76 ± 0.53 | 86.42 ± 0.49 | 91.89 ± 0.32 |
| 19 | Efficient b0 [57] | 93.69 ± 0.45 | 90.26 ± 0.39 | 93.22 ± 0.28 |

We performed a non-systematic review of the literature on methods related to the automated classification of MS, are summarized in Table 5. Most of the methods in the literature relied on deep learning, especially convolutional neural network (CNN) models, to attain high classification results. In contrast, we presented a handcrafted feature-engineering machine learning model for detecting MS on brain MRI. Our ExMPLPQ model attained over 97% binary classification accuracy on all three study data subsets. Only Wang et al. [58] attained a higher classification performance than our model, but their dataset had fewer subjects. In addition, they applied data augmentation on their dataset and used a deep learning model to attain the high classification performance [58], which increased the model's computational complexity. In contrast, our ExMPLPQ algorithm achieved high classification performance with low computational complexity.

Table 6 tabulated that our model attained superior classification results than other previously presented state-of-art methods. The salient points of the proposed ExMPLPQ algorithm are discussed in below.

**Table 6.** Comparison with other state-of-the-art MS brain MRI classification methods.

| Study | Method | Dataset | Subjects | Results, % |
|---|---|---|---|---|
| Plati et al. [59] | Typographic error-based feature extraction, oversampling based feature selection and classification | 78 records, 51 with EDSS 0–3.5, 18 with EDSS 4.0–5.0, 10 with EDSS 5.5–10.0 | 30 MS | Accuracy 94.87, *TP rate for low class 90.40, *TP rate for medium-class 94.20, *TP rate for high class 100.00 |
| Wang et al. [58] | CNN with 14 layers | eHealth Lab and clinic | 38 MS, 26 healthy | Accuracy 98.77, Sensitivity 98.77, Specificity 98.76 |
| Eitel et al. [60] | CNN pretrained on Alzheimer's neuroimaging initiative dataset | Clinic | 76 MS, 71 healthy | Accuracy 87.04, AUC 96.08 |
| Calimeri et al. [61] | Graph neural network | Clinic | 90 MS | Specificity 82, F1-Score 80 |
| Marzullo [62] | Graph CNN | Clinic | 90 MS | Specificity 92, F1-Score 92 |
| Our model | ExMPLPQ | Clinic | 72 MS, 59 healthy | Axial: Accuracy 98.37, Sensitivity 96.46, Specificity 99.60 Sagittal: Accuracy 97.75, Sensitivity 95.01. Specificity 99.80 Hybrid: Accuracy 98.22, Sensitivity 96.39, Specificity 99.50 |

*TP: true positive rate.

Benefits:

- A new brain MRI dataset comprising three study data subsets was prospectively acquired to train and test the model. This dataset has been made publicly available.
- By design, the ExMPLPQ model exploited the advantages of both exemplar and multiple parameters for feature extraction.
- The best features were automatically selected for each of the three binary classification problems using INCA.
- ExMPLPQ attained over 97% accuracies for all study data subsets.
- ExMPLPQ attained better classification performance compared with 19-pre-trained CNNs. Of note, more than a million parameters were required to be assigned/optimized using the deep learning models, which increased their time complexity considerably.
- The ExMPLPQ algorithm has a low time complexity of approximately $O(n \log n)$.
- The base architecture of ExMPLPQ is parametric and is amenable to modification and optimization to create new models using variable patch sizes and updating of feature extractors, feature selectors, and classifiers.

Limitations:

- The dataset was new, which precluded direct comparison with extant methods in the literature. Nevertheless, the common dataset was used to test the ExMPLPQ model and 19 comparator deep learning techniques, which demonstrated superior results for our model.
- Only MS patients admitted to one hospital during one year (2021) were included in the study.
- Patients with less than 9 MS lesions on brain MRIs were excluded from the study.
- Patients with poor MRI image quality and motion artifacts were excluded from the study.

- Patients under 18 years of age were excluded from the study.

## 6. Conclusions

In this research we show that a handcrafted computer vision model is highly accurate for detecting MS based on brain MRI images. The proposed model used LPQ with three $3 \times 3$, $5 \times 5$, and $7 \times 7$ overlapping blocks to extract features from resized brain images and fixed-size patches. The proposed ExMPLPQ was able to detect MS plaques from brain MRI with high accuracy automatically and is computationally lightweight. It has the potential to be implemented in clinics where it supports high-throughput screening of brain MRI images in suspected MS cases. In future works, we hope to acquire larger brain MRI datasets to train and test our model, including MRIs from patients with other diseases, e.g., migraine and vasculitis, that may mimic MS. In addition, more efficient deep learning and handcrafted approaches can be combined with the current model, resulting in more effective learning.

**Author Contributions:** Conceptualization, G.M., B.T., I.T., O.F., P.D.B., S.D., T.T., R.-S.T. and U.R.A.; methodology, G.M., B.T., I.T., O.F., P.D.B., S.D., T.T., R.-S.T. and U.R.A.; software, S.D. and T.T.; validation, G.M., B.T., I.T., O.F., P.D.B., S.D. and T.T.; formal analysis, G.M., B.T., I.T., O.F., P.D.B., S.D. and T.T.; investigation, G.M., B.T., I.T., O.F., P.D.B., S.D., T.T., R.-S.T. and U.R.A.; resources, G.M., B.T., I.T. and O.F.; data curation, G.M., B.T., I.T. and O.F.; writing—original draft preparation, G.M., B.T., I.T., O.F., P.D.B., S.D., T.T., R.-S.T. and U.R.A.; writing—review and editing, G.M., B.T., I.T., O.F., P.D.B., S.D., T.T., R.-S.T. and U.R.A.; visualization, G.M., B.T. and I.T.; supervision, U.R.A.; project administration, U.R.A. All authors have read and agreed to the published version of the manuscript.

**Funding:** This research received no external funding.

**Institutional Review Board Statement:** This research has been approved on ethical grounds by the Non-Interventional Research Ethics Board Decisions, Turgut Ozal University on 3 March 2022 (2022/03).

**Informed Consent Statement:** Not applicable.

**Data Availability Statement:** Not applicable.

**Conflicts of Interest:** The authors declare no conflict of interest.

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
