# Peer review of "An Accurate Multiple Sclerosis Detection Model Based on Exemplar Multiple Parameters Local Phase Quantization: ExMPLPQ"

_applsci, doi:10.3390/app12104920_

Round 1

Reviewer 1 Report

The authors proposed a computationally lightweight machine learning model with handcrafted features for diagnosing Multiple sclerosis on brain images.

The proposed method sounds. However, the following point should be enhanced:
- Validation on the classification was done using 10foldCV, however no standard deviation is reported. In the k-fold CV results, the standard deviation should be reported.
- Are the pretrained models used for comparison fine-tuned on the test datasets? The experiments on pre-trained models should be made fine-tuning the model to better adapt them to the target data. 
- method section: step 8 and step 9 are not clear, especially the connection between equations (13) and (14). Please describe better these points.
Minor issues:

-  The ILPQ acronym seems not be defined.
- abstract: "[...] have been presented in the literature" the final dot is missing
- introduction:"higher accuracy as it collects more information [18]" the final dot is missing
- introduction: Several "[missing reference]" tags are present.
- introduction" "With our work on an accurate MS detection model, we attempt to address some of the issues and problems raised above." you should be more specific about the problems addressed.
- table 1: the caption and the meaning of the * is not clear.
- figure 2: the role of MPLPQ is not clear in the proposed ExMPLQ scheme
- discussion: "we proposed a computationally lightweight machine learning model with handcrafted feature engineering for diagnosing MS on FLAIR brain images, which incorporates a new deep transfer learning framework" what is the new deep transfer learning framework??

Reviewer 2 Report

This paper proposed a method to detect MS disease in MRI scans. It uses deep learning-based feature extraction and then traditional classifiers to detect MS and non-MS classes. The paper is well-written and the results sound. Although the contribution of the paper in the methodology part is limited, introducing a new MS dataset is worthy. I have some minor comments to improve the quality of the paper:

  • The public link of the dataset is not working. You can use repositories like GitHub to publish the dataset.
  • In conclusion, the authors should mention the pros and cons and future works related to their proposed method.
  • It is recommended to write a pseudocode for the algorithm, so the input parameters, steps, and outputs will be easily understood.
  • When comparing with other works, it is important to consider the same experiments and datasets. So, I believe the comparison in Table III is just a review of methods, not a comparison study.

Reviewer 3 Report

The authors performed a diagnostic accuracy study of a "computationally machine learning model for MS diagnosis ". This model offers important and novel characteristic: lightweight, the possibility to run on common desktop and good diagnostic performance.

The paper is interesting, the work is novel, but I have some minor and major concern.

Minor:

  • The abstract should be adapted to journal standard, without caption
  • there are some English grammar and punctation error

Major:

  • in the introduction it should be clarified that 2017 revised Mc Donald criteria, permit to demonstrate dissemination in space and time according to clinical, MRI and CSF data (you stated that VEP and SEP are used to diagnose MS, but the current criteria use clinical, MRI o CSF data to demonstate dissemination in space and time). Please clarify the the characteristics of typical MS lesions on MRI lesions are described in Filippi M et al  BRAIN 2019: 142; 1858–1875 .
  • the term "non-diseased male and female patients" is misleading. please better define the group . are they healthy control?? did you include other neurological disease??  MS lesions needs to be differentiated from many other white matter disease.
  • please better clarify in materials and methods the study design
  • please better clarify in materials and methods the diagnostic gold standard, that you used for send, spec, etc. calculation

Round 2

Reviewer 1 Report

The following sentence is not clear and should be corrected: "[...]where final represents the selected best feature vector; mini. The minimum loss values; 215
and idx, are the indexes of these (minimum) values."

Furthermore:
"The datasets in the studies presented above [22, 28-30] are small." please quantify the word "small".

I suggest a further rereading of the entire work for minor corrections.

Once these things are corrected, in my opinion the manuscript can be accepted.

Reviewer 3 Report

Great Job! 

all my concerns have been clarified.
